# Life Cycle Analysis in the Framework of Agricultural Strategic Development Planning in the Balkan Region

**Michail Tsangas [1], Ifigeneia Gavriel [1], Maria Doula [2] , Flouris Xeni [1] and Antonis A. Zorpas [1,*]**

[1]    Laboratory of Chemical Engineering and Engineering Sustainability, Faculty of Pure and Applied Sciences, Environmental Conservation and Management, Open University of Cyprus, P.O. Box 12794, 2252 Nicosia, Cyprus; tsangasm@cytanet.com.cy (M.T.); ifigenia.gavriel@ouc.ac.cy (I.G.); flouris_xeni@yahoo.com (F.X.)

[2]    Benaki Phytopathological Institute, 8 Stef. Delta, 14561 Kifissia, Greece; mdoula@otenet.gr

*    Correspondence: antoniszorpas@yahoo.com or antonis.zorpas@ouc.ac.cy

**Abstract:** Agricultural sector should be considered, as one of the main economic development sectors in the entire world, while at the same time is responsible for important pollution. The life cycle assessment (LCA) procedure was involved in the agricultural strategic development planning for Balkan region, as a useful tool to identify and quantify potential environmental impacts from the production of apple juice, wine and pepper pesto in three selected sites in Greece, North Macedonia and Bulgaria. These three products were chosen, as are considered as the main economic activities at the areas. The LCA approach covered the entire production line of each product. Based on the LCA results, which comprise the size of six impact categories characterization factors, suggestions were made in order to minimize the footprint of the apples orchard, vineyard and pepper cultivation plots as well as of the production processes of apple juice, wine and pepper pesto as final distribution products. The results indicate that changes in the cultivation and the production must be considered in order to optimize the environmental footprint. Moreover, the whole approach could be useful for agricultural stakeholders, policy makers and producers, in order to improve their products ecological performance, reduce food loss and food waste and increase the productivity of the agricultural sector, while at the same time can improve the three pillars of sustainability through strategy development.

**Keywords:** LCA; agriculture; strategic planning; Balkan region; agri-food

## 1. Introduction

The agricultural sector should be considered as one of the most vital sectors for economic expansion in the world, but, above all, it has an important and crucial role for the food safety, while at the same time is responsible for significant environmental issues such as greenhouse gas (GHG) emissions. The agricultural sector in most countries represents a large percentage of total manufacturing benefit, provides large employers and has significant importance for the gross domestic product (GDP). Nonetheless, this sector is facing challenges, that are mostly related to social issues (e.g., food security control, the rising demand of food, commercial margins), as well as to environmental issues such as climate change, environmental protection and legislation [1,2]. Taking this into consideration, the agriculture sector designs, adopts and implements several new strategies that allow it to face many of these challenges, in order to be productive, sustainable and remain competitive in the marketplace [3,4]. Moreover, the agri-sector (including production and distribution) has been highly influenced by the technological developments in manufacturing methods, currently called "Industry 4.0" [5]. Therefore, these strategies are focused on providing methods, practices and tools that will mainly control and minimize the environmental footprint. Furthermore, the agri-sector plays significant role at the safety of foods, while at the same time is considered as the largest employer on the planet, covering more or

less 40% of the worldwide population. Additionally, it is the biggest source of income and employment in many areas and more specifically for poor rural areas. There are about 500 million small farmers globally according to United Nations (UN), producing more or less 80% of the food consumed, mostly in the development countries. At the same time, since the early twentieth century, 75% of all the crop diversity has been totally lost at all the agriculturalist fields [6].

Agriculture and connected agri-food processing (e.g., cleaning, sorting, juicing, cooking, fermentation, packing, bottling, storage and distribution) cause several environmental impacts such as greenhouse gas (GHG) emissions, water consumption, contamination and deforestation. The size of these impacts may depend on the applied agricultural approaches and practices [7]. Moreover, these are depended on the effective management of fertilizer use and ecosystem services (i.e., use of nutrients and water, soil fertility maintenance and livestock production sustainability) [8]. Increased application of fertilizer nitrogen may improve food production, but due to excessive or unreasonable use it also results to serious environmental issues such as global warming, air pollution, water quality degradation and soil acidification [9]. Agricultural production and harvesting are also one of the common sources of residues of pesticides in food, which contribute to human exposure [10]. Additionally, the agricultural sector contributes to air quality in terms of fine particulate matter pollution and the connected effects on the human health, with the Balkan countries to be in the top list of affected, among the EU-28 [11]. Furthermore, the majority of the anthropogenic methane and nitrous-oxide emissions are produced from the agri-sector [12]. While trends in fuels and electricity consumption in crop production affect the life cycle impacts on the end-user [13], the agri-sector also contributes to the production of GHG affecting the climate change [14]. In this framework, UN sustainable development goals (SDGs) achievement requires sustainable agricultural production. More specifically, SDG 2 focuses on zero hunger, comprises a target to double, by 2030, the productivity of agriculture and the incomes of small-scale producers of food [6], while SDG 6, referring to the clean water and sanitation, includes a target to improve water quality by minimizing release of hazardous substances and materials and reducing pollution by 2030 as well [15]. Both these SDGs aim to duplicate the agricultural productivity and incomes in the small-scale food productions (i.e., Balkan Region) and increase the knowledge and the knowhow of the most sustainable practices, targeting at the same time the protection of ecosystems and to introduce measures to certify the suitable functioning of food commodity markets. Besides, taking into account that SDG 12 focuses on the implementation of reduce and reuse activities to minimize the waste generation and mostly organics and food waste, the agri-sector needs further consideration. According to Zorpas [16] the European Green Deal Strategy has a clear vision of how to achieve climate neutrality by 2050 and aim to implement United Nation Development Program (UNDP) and SDGs by 2030, as well as all EU actions and policies should pull together to help the EU achieve a successful and just transition towards a sustainable future.

Nevertheless, organic agriculture is not the panacea when careful, sustainably run, conventional activities can avoid many environmental impacts connected to high-input agriculture, especially when considering the growing population food needs [17]. Taking this into consideration the knowledge, as well the assessment of the impacts throughout the life cycle of an agriculture product, enables the more effective choice and the improvement of the cultivation as well as of the production methods, in order to achieve their mitigation, in the aim of environmentally effective and sustainable agriculture and agri-food production.

Several tools have been used to assess agriculture sustainability. For example, a multi-level indicator system with adjustable parameters has been applied for smallholder farmland systems [18]. Multi-criteria analysis was adopted for assessing sustainability of agricultural production at the regional level [19]. Furthermore, monetary evaluation has been another approach [20,21]. Nevertheless, agricultural production is the hotspot in the life cycle of food products and LCA can assist to identify more sustainable options [22]. This tool is increasingly used for the improvement of the environmental performance of quarries [23], goods and services, including products belonging to the agri-food

sector [22,24] and it is considered as one of the most important tools for environmental impact assessment [25].

LCA is broadly implemented for agriculture and agri-food products environmental analyses and improvement. It has been used to evaluate the use of energy and the related environmental impacts of pistachio cultivation in Aegina Island in Greece, where the current production and two alternative scenarios were investigated and improvement opportunities were detected [26]. Comparative LCA of three water intensive tree cultivation systems (i.e., pistachio, almond and apple) has identified the "hot-spots" for the crops, exhibiting the most significant environmental impacts and consumption of energy. Moreover, sensitivity analysis was performed in order to investigate actions for water requirements reduction and energy conservation promotion [27]. Furthermore, it has been used to compare different cultivation methods of the same agri-product. For example, Longo et al. [28] applied the methodology to examine the supply chain of organic and conventional apples, including raw materials and energy sources input, the farming step, the post-harvest processes and the apples distribution to the final users evaluating which of the two products is better from an energy and environmental point of view.

LCA has also been used to propose an improved solution to reduce $CO_2$ emissions of wine production in Italy, taking into consideration vinification, bottling, packaging, distribution and waste disposal treatments [29]. The method can also be used as a tool to address sustainable production and consumption patterns of local policies, and to get knowledge for environmental assessment of a wide agricultural production area [30]. Furthermore, it is a suitable method to evaluate how the management of an agri-product growing in a well-defined farming system (e.g., for apples production) influences environmental impacts [31]. It also has been used for managerial decisions justification within agricultural holdings, in order to develop them sustainably in a case study for Romanian viticulture [32]. Additionally, a joint application of the method with data envelopment analysis has been implemented to assess the eco-efficiency of intensive rice production in Japan [33].

Sustainable agriculture performance differs within European Countries and Balkan Countries are included in the low performance cluster. This seems to happen because especially for the Balkan Region huge and strict investments, which are not affordable, are needed. The EU regulations harmonization is made too literally, the population is ageing, as well as there are not proper incentives for younger people to join the profession, and the very producers prefer to operate in grey economy. Moreover, there is lack of properly educated and trained agriculture professionals [34]. Negative impact of agriculture on the environment is dampened by rising income levels [35]. Since gross domestic product (GDP) per capita is lower in Balkan Countries (e.g., Albania, North Macedonia, Serbia, Bulgaria and Romania) comparing to other Europe [36] this is also a weakness for the region.

This paper summarizes the LCA of three pilot sites, of three different crops and the associated processing units, located at three Balkan Countries (i.e., Greece, North Macedonia and Bulgaria). It presents the agriculture and agri-food production system of all stages of the products' life cradle (field to market) (i.e., Thessaloniki port, the life cycle inventory, and chosen life cycle impacts assessment for the three pilot sites).

## 2. Materials and Methods

### 2.1. Method Description

LCA is a standardized procedure that addresses the environmental aspects and potential environmental impacts (e.g., use of resources and the environmental consequences of releases) throughout a product's life cycle from raw material acquisition through production, use, end-of-life treatment, recycling and final disposal (i.e., cradle-to-grave) [37]. It has been developed fast over the past three decades from simple energy analysis to complete life cycle impact assessment, life cycle costing and social-LCA and recently to a more comprehensive life cycle sustainability analysis, which broads the traditional environmental evaluation [23,38]. It is directly applicable for product

development and improvement, strategic planning, evaluation of environmental performance, public policy making, marketing and other [37].

According to ISO 14040:2006, four phases should be followed during the LCA study. The goal and scope definition phase in which the functional unit (FU), studied system boundaries and level of detail of the LCA are specified. The life cycle inventory (LCI) analysis phase which involves the collection of the necessary input/output data with respect to the studied system, in order to meet the goals of the specific study. The life cycle impact assessment phase (LCIA) with purpose to provide information for LCI results of a product's system by assessing the impacts in order to understand their environmental importance and the life cycle interpretation phase at which, the results of the inventory and impact assessment phase are summarized and discussed and the conclusions, recommendations in accordance with the goal and scope, are formed [37].

*2.2. Goal and Scope*

In order to consider all agri-food production processes from the raw material extraction (i.e., water, fertilizers, fuel etc.) to the post-harvest activities and the point where the products are disposed, the system under study included the stages from farm supplies to the distribution, as presented in Figure 1.

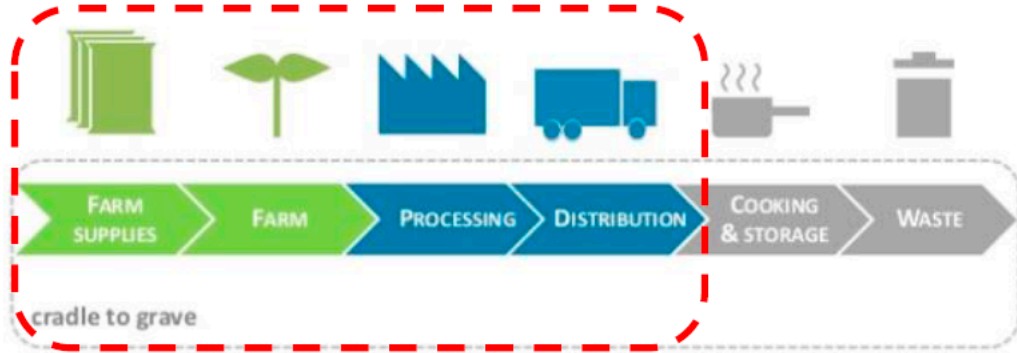

**Figure 1.** System boundaries [39].

The three pilot sites, for which the LCA study was performed, were an apples orchard, a vineyard and a pepper cultivation plot. In order to include the agri-product processing phase as well the relevant environmental impacts and to apply to products that are delivered to the market, the LCA was extended to the apple juice production, the winery and the pepper pesto production. For reliable comparison purposes, the distribution stage was selected to include the transportation to the port of Thessaloniki, as common final destination for all products under study.

Since agriculture production is not stable in the time, the data that were used were extrapolated from the last five years where possible. Moreover, ingredients of the products that are not produced in the pilot farms such as salt and oil for pepper pesto was assumed that are not included in the LCA model.

*2.3. Description of Study Areas*

The pilot sites taken into account for the LCA, were located in three different Balkan Countries as exposed in Figure 2. Specifically, they were a vineyard in Greece, an apple orchard in Bulgaria and a pepper cultivation plot in North Macedonia.

The Greek study farm is located 3 km north of Naousa, which is a city 90 km (more or less) east of Thessaloniki, with approximately 500 ha of cultivated wine growing land and nearby 20 wineries in the wider area. It is a vineyard covering a total surface of 58 hectares, lying at an altitude of 280 to 330 meters, which is the heist point of the Naousa Protected Designation of Origin wine-producing zone. It is planted with Xinomavro (50%), Syrah (15%), Merlot (20%) and Cabernet Sauvignon (10%),

while the rest of the area is covered with various experimental varieties. A winery is established in the vineyard area.

The Bulgarian apple orchard is located in Kustendil District, between the villages of Piperkov Thchiflik, Granitca and Bagrenci. The total orchards area of the pilot farm is 277,606 ha, of which 251,829 ha is covered with apple trees and the rest with plum trees and cherry trees. The factory that produces, among other products, apple juice is located in village Granitsa.

The pepper cultivation farm in North Macedonia is located in the village of Palikura, about 10 km from Kavadarci. At the plot are produced Kapia variety peppers in 0.5 ha in plastic tunnels of 5.1 m × 33 m, but also tomatoes, eggplants and garlics. For crop rotation are applied chickpeas and kidney beans and for weed control are used oats and barley. All harvested crops are transported to a processing facility, which is located in Kavadarci to produce variety of products.

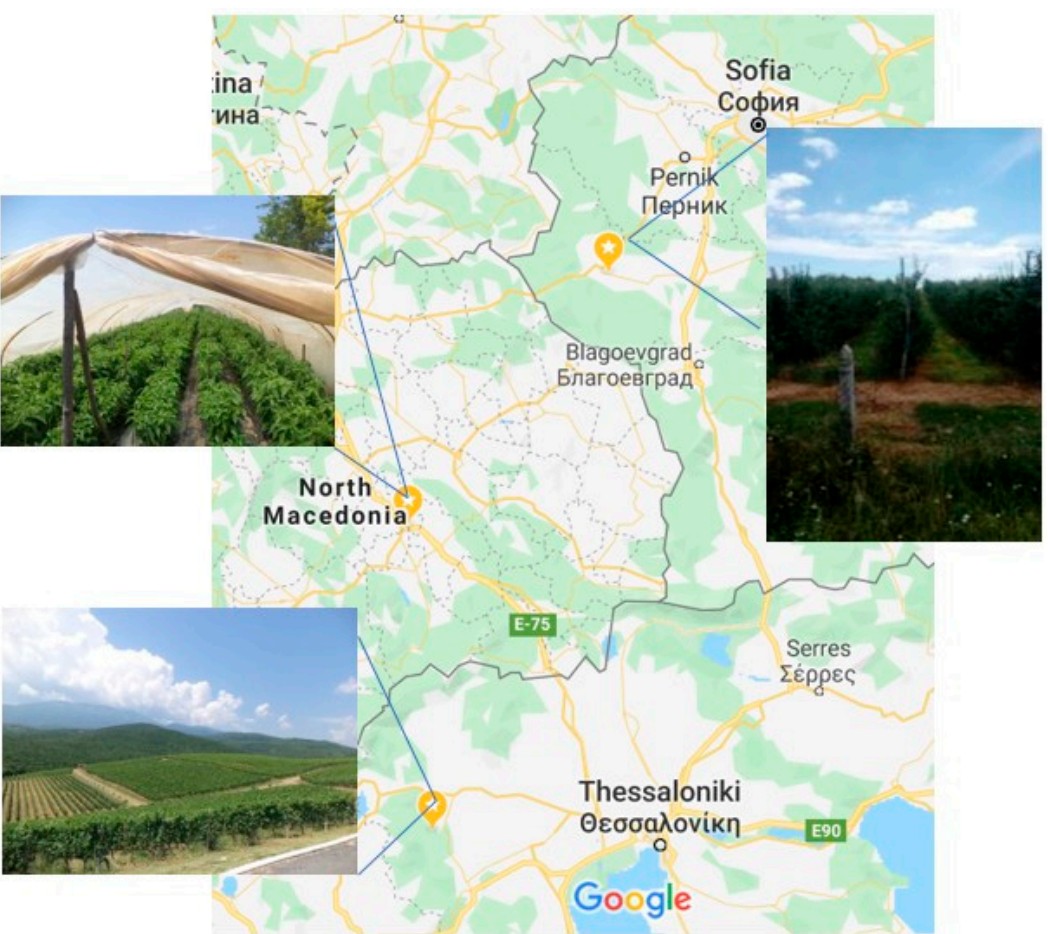

**Figure 2.** The three pilot areas.

## 2.4. Functional Units

The functional unit (FU) shall reflect the marketable product, including expected packaging of the final product. In order to ensure that the input and output data are normalized in a mathematically consistent way, the functional units or/and reference flows shall be clearly determined and be measurable [39,40]. It is important to note that only a product suitable for sale on the market should be taken into account. Table 1, presents the FUs based on the production characteristics for the three pilot areas under study representing the marketable agricultural products for each pilot.

<div align="center">**Table 1.** Functional units.</div>

| Pilot Site | Agricultural Production | Functional Unit | Unit |
|---|---|---|---|
| Greece | Wine/vineyard | One bottle of wine | 0.75 L |
| North Macedonia | Pepper paste/pepper | One jar of pepper "in own sauce" | 0.275 L |
| Bulgaria | Apple juice/apples | One bottle of juice | 1.0 L |

*2.5. Software*

The LCA studies were carried out by using the open source and free software openLCA, which was developed by GreenDelta [41]. Many free and commercial LCA databases and LCIA methods can be imported in the software, so it gives the ability to design a life cycle system by connecting all LCI elements and to quantify the LCIA according to the method used.

*2.6. Data Collection*

The relevant data were collected through in-situ/field campaigns and voluntary survey (taking into account any ethics requirements) of planters, agronomists etc. operating in each pilot area. This approach aimed to increase the credibility of LCA and to draft conclusions according to the local agricultural and economic circumstances [39]. Therefore, primary site-specific data was obtained with the use of a questionnaire (asking data for chemicals, water, energy, material equipment, waste, emissions, by-products and productivity) for each pilot farm. Where possible, based on the primary data derived, the direct emissions in water, soil and air from field activities were estimated. To complete the life cycle inventory, data associated with the activities performed in the background system (i.e., production of agro-chemicals, fertilizers and machinery and transportation) were taken from literature and other available LCI databases (i.e., Ecoinvent v 3.3 and Agribalyse v 1.2 and v 1.3). Collected data included both cultivation and processing/post-harvest activities for agricultural production from the past five years (2013–2017) for the two pilots of Bulgaria and Greece, while for the North Macedonian pilot area, data of the last three years were used.

*2.7. Impact Categories*

The six environmental impact categories (further to the goal and the scope), which were calculated in the LCIA, for the pilot farms under examination were acidification potential (AP) measured in kg $SO_2$-eq·$FU^{-1}$, eutrophication potential (EP) in kg $PO_4$-eq·$FU^{-1}$, global warming potential (100 years) (GWP) in kg $CO_2$-eq·$FU^{-1}$, ozone depletion potential (ODP) in kg CFC-11-eq·$FU^{-1}$, photochemical ozone creation potential (POCP) in kg $C_2H_4$-eq·$FU^{-1}$ and cumulative energy demand (CED) in MJ-eq·$FU^{-1}$. Those impact categories are linked to the main environmental burdens of agriculture production including emissions to soil, air and water, as well as energy consumption.

The first five of them are specified by the CML 2001 (April 2013 and January 2015 version) impact assessment method of the Centre of Environmental Science of Leiden University [42]. The CED impact category as an energy flow indicator can be calculated according to Frischknecht et al. [43].

Moreover, these impact categories were chosen as they have also been used for LCIA for similar to the pilot areas FU, activities or products. Some relevant literature extracted results are presented in Table 2.

**Table 2.** Similar activities Life Cycle Impact Assessment (LCIA) (literature extracted).

| Product—Functional Unit (Reference) | AP (kg SO$_2$-eq·FU$^{-1}$) | EP (kg PO$_4$-eq·FU$^{-1}$) | GWP (kg CO$_2$-eq·FU$^{-1}$) | ODP (kg CFC-11-eq·FU$^{-1}$) | POCP (kg C$_2$H$_4$-eq·FU$^{-1}$) | CED (GJ-eq·FU$^{-1}$) |
|---|---|---|---|---|---|---|
| 0.75 L bottle of organic red wine [24] | 0.013 | 0.006 | 1.704 | $1.43 \times 10^{-7}$ | $5.3 \times 10^{-4}$ | |
| One bottle of red high quality wine [29] | | | 1.58 | $3.91 \times 10^{-8}$ | | |
| One t of fresh apple [27] | 0.95 | 0.44 | 89 | | | 1.21 |
| Apples production ha$^{-1}$ (means) [31] | 25.4 | | 2600 | | | 37.6 |
| Greenhouse peppers (mean of a group of foods/kg produce) [44] | | | 1.02 | | | |
| Pepper (FU 1000 kg) [30] | 6.9 | 3.4 | 915.5 | $4.0 \times 10^{-4}$ | 0.3 | 18 |
| Apples production 1 t (conventional) [28] | | | $6.12 \times 10^2$ | $8.54 \times 10^{-5}$ | | |
| Apples production 1 t (organic) [28] | | | $5.88 \times 10^2$ | $8.46 \times 10^{-5}$ | | |

Acidification Potential (AP), Eutrophication Potential (EP), Global Warming Potential (100 years) (GWP), Ozone Depletion Potential (ODP), Photochemical Ozone Creation Potential (POCP), Cumulative Energy Demand (CED).

## 3. Results

### 3.1. Systems Modelling

Although LCA may vary from industry to industry and for the same products [25], Figure 3 provides the common information regarding the inputs, processes and outputs for the systems that were studied. Justifications for each pilot study area have been done. In regard to cultivation, planting was applied only to the pepper crop, where pruning was not included in the activities. For processing, fermentation was relevant only for wine production. Moreover, the "emissions" outputs included all the emissions to water, soil, air and groundwater.

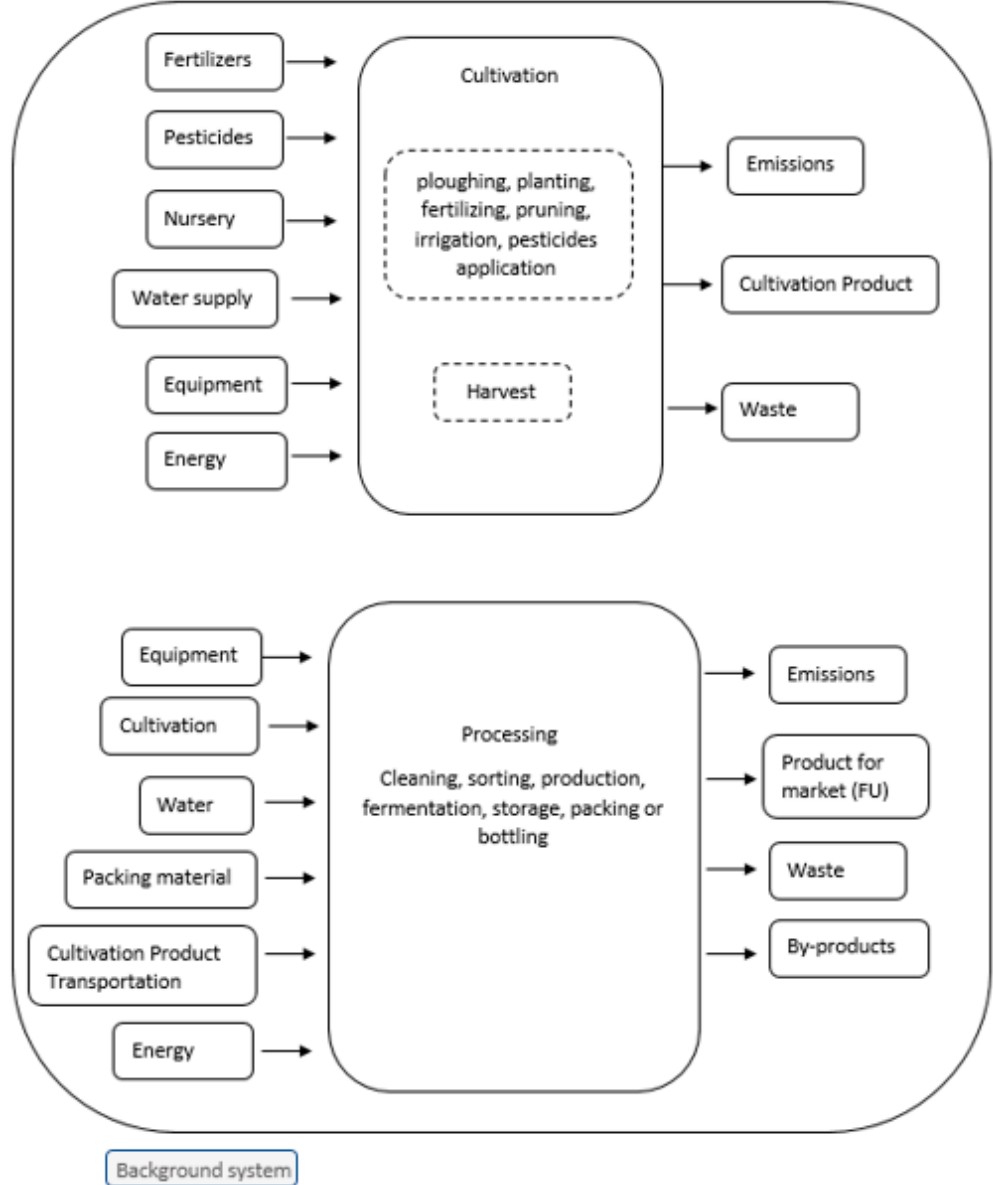

**Figure 3.** Presentation of system inputs, processes and outputs.

### 3.2. Life Cycle Inventory (LCI) Analyses

LCI analyses is an inventory of the input/output data with regard to the system being studied [38]. The primary data elements for the LCI for each pilot area were included in the filled questionnaires and are presented in Table 3.

**Table 3.** Life Cycle Inventory (LCI) primary elements.

| Input/Output | Pilot Area | | |
|---|---|---|---|
| | Greece (Grapes/Wine) | North Macedonia * (Peppers/Pepper Paste) | Bulgaria (Apples/Apple Juice) |
| Orchard production (average last 5 years *) | 8090 kg/ha | 48,000 kg/ha | 1690 kg/ha |
| Fertilizers | ΥARA BELLA (P) 200 kg/ha/y Patenkali (K) 200/ha/3 y | KAN (potassium ammonium nitrate) 300 kg/ha/y NPK 10-30-20 500 kg/ha/y | $NH_4NO_3$- 120 kg/ha/y $P_2O_5$ 80 kg/ha/y $K_2O$ 200 kg/ha/y |
| Organic fertilizers | | 30 manure t/ha/3 y 0.7% N $P_2O_5$ 0.22% K 20 0.8% | |
| Pesticides | 63.75 L/ha/y | 21 kg/ha/y | 5 L/ha/y |
| Irrigation water | 0–1000 m$^3$/ha/y | 30 m$^3$/ha/y | 1400 m$^3$/ha/y |
| Orchard diesel demand | 7000 L/y | 300 L/y | 15,000 L/y |
| Orchard electricity demand | 0 | 0 | 200,000 KWh/y |
| Orchard lubricants demand | 250 L/y | 4 L/y | 50 L/y |
| Waste by orchard | 2800–3000 kg/ha pruning; 7.5 kg/ha plastic containers (recycling); | 15 kg of pesticides packaging/years; 333 kg/ha/y of plastic tunnel film; 1500 kg crop residues/y; 400 kg/ha/y of black plastic mulching film; | 4800 kg/ha pruning; 12 kg/ha plastic containers (recycling); |
| Working hours (cultivation) | 372 working hours/ha/y | 786 working hours/ha/y | 1135 working hours/ha/y |
| Processing water consumption | 0.2 m$^3$/t of processed grape | 4.5 m$^3$/t of processed peppers | n/a |
| Processing electricity demand | 84,000 kWh/y | 190 kWh/month | 40,000 kWh/y |
| Processing gas demand | - | 60 L/t (LPG) | n/a (Natural Gas) |
| Processing lubricants demand | 10 L/y | 20 L/y | 8000 kg/y |
| Packaging material | 1 glass bottle, paper label/FU | 1 glass jar, lid and paper label/FU | 1 glass bottle and PVC label/FU |
| Waste by processing | 20% Grape marc | Organic waste 35 tons/year; plastic 300 kg/y; paper 100 kg/y; | 90% apple jelly; apple skins 1000 kg/ha/y; |
| Waste water by processing | 0.2 m$^3$/t of processed grape | 300 m$^3$/y | n/a |
| Transportation distance (orchard to factory) | 50–500 m | 10 km | 4 km |
| By-products | 20% grape marc | None | Apple flower |
| Equipment (orchard and product preparation) | Cars; agriculture tractors; irrigation pumps; power and hand tools; conveyor;lifts; pallet trucks; freezing chambers; bottling machine; labelling machine; wine pumps; | Agriculture tractors; irrigation pump; freezing chambers; unloader; conveyor; roasting ovens; peeling machine; cutting machine;cooking machine; filling machine; labelling machine; LOT printer; water pumps; lifts; pallet trucks; power and hand tools; pasteurization machine | Cars; agriculture tractors; irrigation pumps; power and hand tools; lifts;freezing chamber; pressing machine; pasteurization machine; mimmer machine; bottling and packing machine; water pumps; |
| Functional Unit production | 0.6 L wine/kg grapes | 1.5 jars/kg pepper | 0.6 L juice/kg apples |

* For North Macedonia reference to last three years.

Secondary data regarding orchard and processing, as well as the background system, became available through data sources as relevant scientific literature and databases (i.e., Eco-invent and Agribalyse).

### 3.3. Data Quality

In order to collect and assess all the primary data used for LCA studies (and for all the examined areas), in-situ surveys were used, while secondary data were drawn from well-established LCI databases (i.e., Ecoinvent v3.3 and Agribalyse). An assessment of the quality of the data (Table 4) for the LCI for the three pilot areas has been prepared according to the Ecoinvent guidelines [45].

**Table 4.** Life Cycle Inventory quality assessment.

| Assessment Indicator | Indicator Score Table 10.4 [45] | |
| --- | --- | --- |
| | Primary Data | Secondary Data |
| Reliability | 3 | 3 |
| Completeness | 1 | 5 |
| Temporal correlation | 1 | 5 |
| Geographical correlation | 1 | 4 |
| Further technological correlation | 1 | 4 |

Although the primary data acquired by the survey are of high quality, the secondary data, mainly concerning the background systems, could be improved. The main issue is that accurate data for Balkan Countries are not included in the available databases. However, this could be solved by the creation of specific local databases, which would include long term observations and information from a wide sample.

### 3.4. Life Cycle Impact Analyses

The life cycle impact categories for the three pilot areas have been calculated using the openLCA software. The CML 2001 (April 2013 version) was used for the Greek pilot area, while the CML (baseline) (v4.4, January 2015) European reference inventories was applied for the other two pilot areas. The calculation of the cumulative energy demand (CED) impact category was based on the method proposed by [43].

The impacts of each category as calculated for the three pilot areas are presented in Table 5.

**Table 5.** Life Cycle Impact Assessment for the three pilot areas.

| Pilot Site (FU) | AP (kg $SO_2$-eq·$FU^{-1}$) | EP (kg $PO_4$-eq·$FU^{-1}$) | GWP (kg $CO_2$-eq·$FU^{-1}$) | ODP (kg CFC-11-eq·$FU^{-1}$) | POCP (kg $C_2H_4$-eq·$FU^{-1}$) | CED (MJ-eq·$FU^{-1}$) |
| --- | --- | --- | --- | --- | --- | --- |
| Greece (one 0.75 L bottle of wine) | $1.98 \times 10^{-2}$ | $5.62 \times 10^{-3}$ | 1.10 | $2.21 \times 10^{-7}$ | $4.48 \times 10^{-4}$ | 21.3 |
| Bulgaria (one 1 L bottle of juice) | 14.1 | 3.49 | $2.02 \times 10^3$ | 0.00 | $1.26 \times 10^{-1}$ | $2.27 \times 10^6$ |
| North Macedonia (one 275 mL jar of pepper paste in own sauce) | $3.50 \times 10^2$ | $1.37 \times 10^2$ | $7.46 \times 10^4$ | $3.63 \times 10^{-3}$ | $2.27 \times 10^6$ | $7.13 \times 10^5$ |

Acidification Potential (AP), Eutrophication Potential (EP), Global Warming Potential (100 years) (GWP), Ozone Depletion Potential (ODP), Photochemical Ozone Creation Potential (POCP), Cumulative Energy Demand (CED).

## 4. Discussion

Therefore, from the environmental perspective, as the agri-sector participates in the production of food, it has several environmental impacts. Typically, if food is lost or wasted, this implies poor exploitation of resources. The agri-sector must increase its productivity to cover the needs of the increasing population by 35%–55% according to FAO [46], since 2012 to 2050. This highlights the need to control food loss and waste in order the resource use efficiency to be improved. In terms of food loss

FAO indicated that the highest level of loss obtained from food groups and roots (26%), followed by fruit and vegetables cultivation (22.5%).

Although, LCA and LCIA are widely used in studies to assess the life cycle environmental impact of several agriculture and agri-food products including the pilot farms products [24,27–29,31,44,47], the existing literature does not cover neither the total range of impact selected categories to be assessed, nor all the products under study. Nevertheless, through the LCA and LCIA approaches useful information, related with products footprint, could be received. It is almost true that packaging [46] (among other, LCA provides data regarding the packaging) can decrease food loss and control food waste, and its related environmental impacts on water and land use, as well as on GHGs emissions (despite the fact that packaging also contributes to GHGs emissions and increases the use of plastics). Moreover, LCA provides data related to food loss and environmental impacts of the food supply chain, although the various stages of it differ from country to country and from area to area, mainly due to economic dimensions [46]. For example, the carbon footprint of food loss and waste tracks a norm throughout the numerous food supply chain phases, rather diverse from that of land or water footprints. Therefore, the reduction measures that aim to decrease the carbon footprint should not be similar to those pointing to decrease scarcity of water or degradation of land.

The life cycle of wine in a 0.75 L glass bottle has been analyzed for several production areas and varieties; however, there are no (to the best of our knowledge) LCA studies for the production of pepper pesto and apple juice. Therefore, the literature available information for the five mid-point environmental impact categories (i.e., AP, EP, GWP, ODP, POCP and CED) do not cover the range selected for the pilot orchards and products. Literature-extracted LCIA [24,29] could help for some conclusions, but such an interpretation is not able to provide all the needed information for further use towards farms with zero carbon-, waste- and water-footprint.

Nevertheless, the values of the five impact categories (i.e., AP, EP, GWP, ODP and POCP) per one 0.75 l bottle of wine according to Arzoumanidis et al. and Iannone et al. [24,29] are comparable with the values extracted by the LCA performed for the Balkan Region within this paper (i.e., AP ($1.98 \times 10^{-2}$ kg $SO_2$-eq·$FU^{-1}$), EP ($5.62 \times 10^{-3}$ kg $PO_4$-eq·$FU^{-1}$), GWP ($1.10$ kg $CO_2$-eq·$FU^{-1}$), ODP ($2.2198 \times 10^{-7}$ kg CFC-11-eq·$FU^{-1}$), POCP ($4.48 \times 10^{-4}$ kg $C_2H_4$-eq·$FU^1$) and CED ($21.3$ MJ-eq·$FU^{-1}$)). This verifies the consistency of the work method was followed for the present research. Moreover, it is observed that the Greek pilot GWP and POCP values are quite lower than those available from the one literature study [24], EP value is close to results of the same work [24] and ODP value lies between the two above mentioned available LCIA results [24,29]. The only impact category that has a quite higher value for the Balkan region pilot is the AP, which was $1.98 \times 10^{-2}$ kg SO2-eq per one 0.75 l bottle of wine.

The interpretation of the LCI and LCIA of the three pilot farms illustrates the differences of agriculture as well as processing practices followed in each one. Taking into consideration that three different crops and agri-food products are studied, the differences of inputs and outputs size between the pilot areas feature issues could be discussed, but impact categories size cannot be compared. The different needs of material, land, chemicals, energy, equipment and different productivities and waste production of the three cultivations cause non-comparable sizes of environmental impacts, depending on the special characteristics of each. However, a common observation is that the five selected environmental impact categories have value for all the pilot areas (except ODP for the Bulgarian pilot), so the potential harm to the environment of all the three of them is significant. Therefore, selected measures (i.e., energy consumption, material, water and chemicals use and waste production minimization as well as increase of environmentally friendly methods and material use) should and could be implemented to mitigate them.

According to Guinée [42] for the AP impact category, the major acidifying pollutants are considered to be $SO_2$, $NO_x$ and $NH_x$. For EP the excessively high environmental levels of macronutrients such as nitrogen (N) and phosphorus (P) are important. For ODP, emissions of CFC and Halon are a major issue. Furthermore, toluene, trans–2-Butene, trans–2-Hexene and trans–2-Pentene, among others,

are significant for POCP. GWP is dependent on GHG emissions to the air throughout the full process. Additionally, all life cycle energy needs and energy intensive factors affect the CED of each product.

Measures to decrease either use or emission of these elements could contribute to impact categories size. Higher use of fertilizer is not necessarily connected to larger crop yields [8,9]. Therefore, as this is related to N and P, as well as energy use, reduced application could contribute to impact categories size mitigation, without significant influence on production volume. Moreover, technology implementation to improve agriculture efficiency can help to reduce emissions [11]. Renewable energy use could also be an option to minimize the agricultural sector impacts on the environment [48]. Transition from conventional to organic agricultural production could also be an option [12], although this is not considered to absolutely improve environmental performance of the sector [17,49].

## 5. Conclusions

In order the target of farms with zero carbon-, waste- and water-footprint to be achieved, good agriculture as well as production practices should be obvious throughout the orchard to market life of all the three products. Depending on the crop and product processing, issues to consider, in order to mitigate the environmental impacts of the three pilot areas activities, are the following:

- The use of chemical fertilizers and pesticides as well as the waste plastic film should be minimized, and products with low life cycle environmental impact should be preferred.
- The use of packaging materials should also be minimized and raw material with low life cycle environmental impact should also be preferred.
- Renewable energy should be preferred in all product processing phases.
- Equipment with low life cycle environmental impact should be used. Equipment that is used in all phases should be well maintained.
- Fruits or vegetables and final product cooling equipment should be controlled and well maintained.
- Low environmental impact transportation means should be preferred.
- Transportation distance should be minimized where possible.
- Waste reducing, reusing and recycling actions should be adopted in all life cycle of the pilot products.

However, some of the above suggestions could be prioritized or localized. The transportation distances at North Macedonian and Bulgarian sites are long and they should be reduced, when agricultural trucks should not be used for products transportation and be substituted by trucks. Moreover, renewable energy production (e.g., by local photovoltaic systems establishment) could immediately reduce the energy connected environmental impacts. As soon as cooling equipment is necessary for all the three products, this would minimize its effect as well. Furthermore, Greek and Bulgarian sites could also consider the use of organic fertilizers. Concerning the use of equipment and material with low life cycle environmental impact, this shall be necessary for any new or replacements for all three sites.

Concluding, through the LCA approach, it can be mentioned that any of the above suggestions could be used in any of the proposed pilot areas and beyond in the Balkan Region, in order to control, monitor and assess the environmental performance, from the cultivation to the distribution, promoting at the same time a strategic line for the agricultural sector.

**Author Contributions:** Conceptualization, A.A.Z. and M.D.; methodology, A.A.Z., M.D. and M.T.; validation, I.G., M.T. and A.A.Z.; data curation, I.G., M.T. and F.X.; writing—original draft preparation, M.T.; writing—review and editing, A.A.Z. and M.T.; supervision, A.A.Z.; project administration, I.G. All authors have read and agreed to the published version of the manuscript.

**Funding:** This research was funded by Interreg Balkan Mediterranean, under the BalkanRoad project, towards farms with zero carbon-, waste- and water-footprint. Roadmap for sustainable management strategies for Balkan agricultural sector. Interreg: 2432 (grant number).

**Acknowledgments:** The authors wish to thank the owners and personnel of the pilot sites as well as the project partners for their help and contribution.

**Conflicts of Interest:** The authors declare no conflicts of interest.

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
