# Peer review of "Life Cycle Analysis in the Framework of Agricultural Strategic Development Planning in the Balkan Region"

_sustainability, doi:10.3390/su12051813_

Round 1

Reviewer 1 Report

The topic is interesting, methodology is generally correct, and results are of practical relevance.

The impact of cultivation on groundwater (nitrates) is not considered, or it is not explicitly included in the “emission” output (Fig 3). Please clarify this point, either in figure or in the text/tables. The scores of LCI assessment indicators for secondary data are in the range 3-5 (Table 5), indicating “low quality data” according to Ecoinvent guidelines. Authors state that they “could be improved” (line 253): could you please indicate how? In the conclusion section, there is a list of findings (lines 336-348). Of course these 8 statements are all correct and supported by data, however they are quite obvious since they mainly rely on well-known best practices, and a detailed LCA analysis is not necessary to provide these indications. Authors should instead use their results for indicating which of these 8 actions are prioritary. For example, in the case of transportation, which action would be more effective to cut CO2 emissions: reducing distance or improving the transportation infrastructure? Lines 120-142 contain a general description of LCA methodology, which is not necessary: they should be reduced It is not clear why Table 3 is in the Materials and Methods section: it is a list of results obtained from similar studies, it can be moved to Introdicution. Otherwise, it can be used for a comparison with results from this case study, in the Discussion section.

Minor comments

Resolution of figure 1 is not acceptable Table 1: please use the same unit (l or ml)

Author Response

We would like to thank the reviewer for the constructive comments and suggestions. Please find below our response.

The impact of cultivation on groundwater (nitrates) is not considered, or it is not explicitly included in the “emission” output (Fig 3). Please clarify this point, either in figure or in the text/tables.

This has been clarified in the text.

The scores of LCI assessment indicators for secondary data are in the range 3-5 (Table 5), indicating “low quality data” according to Ecoinvent guidelines. Authors state that they “could be improved” (line 253): could you please indicate how? Recommendation has been added.

In the conclusion section, there is a list of findings (lines 336-348). Of course these 8 statements are all correct and supported by data, however they are quite obvious since they mainly rely on well-known best practices, and a detailed LCA analysis is not necessary to provide these indications. Authors should instead use their results for indicating which of these 8 actions are prioritary. For example, in the case of transportation, which action would be more effective to cut CO2 emissions: reducing distance or improving the transportation infrastructure? A paragraph has been added in conclusions section.

Lines 120-142 contain a general description of LCA methodology, which is not necessary: they should be reduced. The description was reduced.

It is not clear why Table 3 is in the Materials and Methods section: it is a list of results obtained from similar studies, it can be moved to Introduction. Otherwise, it can be used for a comparison with results from this case study, in the Discussion section. Table 3 was included in materials and methods section because it supports the selection of the six impact categories for LCIA, moreover results from this study are compared with these in the table in discussion where this is applicable.

Minor comments

Resolution of figure 1 is not acceptable the figure size was changed in order the resolution to be improved. Table 1: please use the same unit (l or ml) This was done.

Reviewer 2 Report

The paper applies the life cycle analysis to quantify the environmental impacts throughout the life cycle of the agricultural products. The authors seem to apply the analysis technique to analyze the environmental impacts of the agricultural industry in Balkan and develop several policy implications. However, I do not see any new methodological development for such an analysis. Neither do I have any strong data-driven conclusion (statistical evidence) that provides considerable insights for agricultural policy-making (with respect to eco-friendly and sustainable farming). If there are any, please specify the points clearly and provide evidence for the claim.

Author Response

We would like to thank the reviewer for the constructive comments and suggestions. Please find below our response.

The paper applies the life cycle analysis to quantify the environmental impacts throughout the life cycle of the agricultural products. The authors seem to apply the analysis technique to analyze the environmental impacts of the agricultural industry in Balkan and develop several policy implications. However, I do not see any new methodological development for such an analysis. Neither do I have any strong data-driven conclusion (statistical evidence) that provides considerable insights for agricultural policy-making (with respect to eco-friendly and sustainable farming). If there are any, please specify the points clearly and provide evidence for the claim.

An explanation has been added in introduction closing.

Reviewer 3 Report

The Abstract, introduction and study desidn are well described. Also the findings are reasonable and well described. 

The only remark is that the conclusion is quite General and should Focus more precisely on the findings of the study. The authors claim e.g. in the abstract to derive strategic Options and improvements according to the findings of the study. Indeed the authors give strategic Guidelines but hese Guidelines are quite General. More precise "Hands-on" recommendations according to the study results would be helpful like e.g in which Areas could the use of Energy reduced and how (with WHAT alternatives). Not just "Low environmental impact transportation means should be preferred"....plese give an advice how this could be realized and what Impact this will have...

Author Response

We would like to thank the reviewer for the constructive comments and suggestions. Please find below our response.

The only remark is that the conclusion is quite General and should Focus more precisely on the findings of the study. The authors claim e.g. in the abstract to derive strategic Options and improvements according to the findings of the study. Indeed the authors give strategic Guidelines but hese Guidelines are quite General. More precise "Hands-on" recommendations according to the study results would be helpful like e.g in which Areas could the use of Energy reduced and how (with WHAT alternatives). Not just "Low environmental impact transportation means should be preferred"....plese give an advice how this could be realized and what Impact this will have...

A relevant paragraph has been added in conclusions section.

Reviewer 4 Report

Dear Authors,

The article may have interesting implications for business and policy development in Balkan coutries; however, a lot of work is still needed to make the article suitable for publication in a scintific journal.

The abstract is very general; please stick to the aims, results and recommendations of your study.

The introduction is very general as well; please focus on the case study areas and provide strong rationale for their seleciton and representativity. At least you should be able to say what this research adds to the literature; this is critical for article publication. Please add a mini-literature review about recent lcas of the selected supply chains and about recent food/agricultural product lcas in the selected countries. A rationale behind country selection should be provided as well. This is also critical for article publication; in the present draft I don’t understand how the three countries can represent the Balkan region…

The life cycle inventory should be moved to the materials and methods section; emission calculations fro the foreground system are missing from the LCI… References are needed per individual data inputs (including primary data) and emission factors. Figure 3: I can’t see the difference between the background and foreground systems; this picture should be included in the methods as well, besides I think one specific picture per product is needed. Not providing data hinders study reproducibility.

The results section is very poor and contribution analysis is missing, which prevents you to from the hot-spot analysis you anticipated among research aims.

The recommendations arising from your research and, mostly the policy implications that you anticipated in the introduction and abstract, are completely missing.

Author Response

We would like to thank the reviewer for the constructive comments and suggestions. Please find below our response.

The abstract is very general; please stick to the aims, results and recommendations of your study.

The introduction is very general as well; please focus on the case study areas and provide strong rationale for their selection and representativity. At least you should be able to say what this research adds to the literature; this is critical for article publication. The abstract was modified.

Please add a mini-literature review about recent lcas of the selected supply chains and about recent food/agricultural product lcas in the selected countries. A literature review is included in the introduction and additionaly in paragraph 2.7 (to support the selection of the specific impact categories) including the detected lca for Balkan food/agriculture products (Greece & Romania – no other is available to our best knowledge).

A rationale behind country selection should be provided as well. This is also critical for article publication; in the present draft I don’t understand how the three countries can represent the Balkan region… A relevant explanation has been added at the introduction end.

The life cycle inventory should be moved to the materials and methods section; emission calculations fro the foreground system are missing from the LCI… References are needed per individual data inputs (including primary data) and emission factors.

LC Inventory is one of the four phases of the study. That is why is included in results.

Primary data have been derived from site questionnaires and secondary data from the mentioned databases in data collection paragraph (2.6) and in Life inventory paragraph (3.2).

Figure 3: I can’t see the difference between the background and foreground systems; this picture should be included in the methods as well, besides I think one specific picture per product is needed. Not providing data hinders study reproducibility. Background systems are mentioned in paragraph 2.6. (production of agro-chemicals, fertilizers and machinery and transportation)

The results section is very poor and contribution analysis is missing, which prevents you to from the hot-spot analysis you anticipated among research aims. We agree that contribution analysis would help to further and more specific recommendations but this was not prepared for all the three sites.

The recommendations arising from your research and, mostly the policy implications that you anticipated in the introduction and abstract, are completely missing. A paragraph has been added in conclusions section

Round 2

Reviewer 4 Report

Dear Authors,

I don't think the article has changed.

I am forwarding the comments to the previous version.

The article may have interesting implications for business and policy development in Balkan coutries; however, a lot of work is still needed to make the article suitable for publication in a scintific journal.

The abstract is very general; please stick to the aims, results and recommendations of your study.

The introduction is very general as well; please focus on the case study areas and provide strong rationale for their seleciton and representativity. At least you should be able to say what this research adds to the literature; this is critical for article publication. Please add a mini-literature review about recent lcas of the selected supply chains and about recent food/agricultural product lcas in the selected countries. A rationale behind country selection should be provided as well. This is also critical for article publication; in the present draft I don’t understand how the three countries can represent the Balkan region…

The life cycle inventory should be moved to the materials and methods section; emission calculations fro the foreground system are missing from the LCI… References are needed per individual data inputs (including primary data) and emission factors. Figure 3: I can’t see the difference between the background and foreground systems; this picture should be included in the methods as well, besides I think one specific picture per product is needed. Not providing data hinders study reproducibility.

The results section is very poor and contribution analysis is missing, which prevents you to from the hot-spot analysis you anticipated among research aims.

The recommendations arising from your research and, mostly the policy implications that you anticipated in the introduction and abstract, are completely missing.

Author Response

Dear Reviewer 

We thank you for your effort to evaluate the paper and we believe that we have answer all your comments 

We submit with track changes the manuscript again

Also see our comments bellow regarding your comments 

Response to reviewer 4. Round 2

We would like to thank the reviewer for the constructive comments and suggestions. Please find below our response.

1. The abstract is very general; please stick to the aims, results and recommendations of your study.

Answer

The abstract was totally modified. Moreover, In the abstract is mentioned:

the aim is “to identify hot spots of agri-production, where the most significant impacts occur, giving the opportunity to develop strategies to improve the environmental performance of the agricultural sector.”

The results “comprise the size of six impact categories characterization factors”,

Recommendations: “suggestions were made in order to minimize the footprint of the apples orchard, vineyard and pepper cultivation plots as well as for the production of apple juice, winery and pepper pesto. The recommendations covered supplies, equipment and processes and could be useful for policy makers and producers, in order to improve the products ecological performance, reduced the impact of food loss and food waste and increase the agri-sector productivity.”

2. The introduction is very general as well; please focus on the case study areas and provide strong rationale for their selection and representativity. At least you should be able to say what this research adds to the literature; this is critical for article publication.

Answer 

Regarding the research adds to the literature please see above comment regarding results and recommendations.

Furthermore, to case study areas selection and representativity the last paragraph of the introduction was modified: “The results interpretation emerges the environmental issues of the cultivations and since the method enables in-depth analysis and the selected crops cover a range of products and applied process, it can lead to conclusions and recommendations to develop strategies for improving their ecological performance, able to be extended to the total Balkan Region.”

3. Please add a mini-literature review about recent lcas of the selected supply chains and about recent food/agricultural product lcas in the selected countries.

Answer 

A literature review is included in the introduction (see line 80 – 113) and additionally in paragraph 2.7 (Moreover, Table 2 to support the selection of the specific impact categories) including the detected LCA for Balkan food/agriculture products (Greece & Romania – no other is available to our best knowledge).

A rationale behind country selection should be provided as well. This is also critical for article publication; in the present draft I don’t understand how the three countries can represent the Balkan region…

Please see comment on to case study areas selection and representativity: “The results interpretation emerges the environmental issues of the cultivations and since the method enables in-depth analysis and the selected crops cover a range of products and applied process, it can lead to conclusions and recommendations to develop strategies for improving their ecological performance, able to be extended to the total Balkan Region.”

Dear Reviewer the selected areas were chosen due to the fact that this this research was funded by Interreg Balkan Mediterranean, under the BalkanRoad project, towards farms with zero carbon-, waste- and water-footprint. Roadmap for sustainable management strategies for Balkan agricultural sector, and was indicated from the beginning of the project.

3. The life cycle inventory should be moved to the materials and methods section; emission calculations from the foreground system are missing from the LCI… References are needed per individual data inputs (including primary data) and emission factors.

Answer 

LC Inventory is one of the four phases of the study. That is why it was included in results.

Primary data have been derived from site questionnaires and secondary data from the mentioned databases in data collection paragraph (2.6) and in Life inventory paragraph (3.2) i.e. LCI databases Ecoinvent v 3.3 and Agribalyse v 1.2 and v 1.3..

4. Figure 3: I can’t see the difference between the background and foreground systems; this picture should be included in the methods as well, besides I think one specific picture per product is needed. Not providing data hinders study reproducibility.

Answer 

Background systems are mentioned in paragraph 2.6. (production of agro-chemicals, fertilizers and machinery and transportation)

5. The results section is very poor and contribution analysis is missing, which prevents you to from the hot-spot analysis you anticipated among research aims.

Answer

We agree that LCA contribution analysis would help to further and more specific recommendations but this was not prepared for all the three sites. This is a very good remark for further research.

5. The recommendations arising from your research and, mostly the policy implications that you anticipated in the introduction and abstract, are completely missing.

Answer

A paragraph has been added in conclusions section: “However, some of these suggestions could be prioritized or localized. The transportation distances at North Macedonian and Bulgarian site are long and they should be reduced, when agricultural trucks should not be used for products transportation and be substituted by trucks. Moreover, renewable energy production e.g. by local photovoltaic systems establishment, could immediately reduce the energy connected environmental impacts. As soon as cooling equipment is necessary for all the three products, this would minimize its effect as well. Furthermore, Greek and Bulgarian site could also consider the use of organic fertilizers. Concerning the use of equipment and material with low Life Cycle Environmental Impact, this shall be necessary for any new or replacement for all three sites.”

Additionally, the following were already included in the conclusions paragraph initial version:

Chemical fertilizers and pesticides use as well as plastic film waste should be minimized and products with low life cycle environmental impact should be preferred. Packaging materials use should also be minimized and raw material with low life cycle environmental impact should be preferred. Renewable energy should be preferred in all product processing phases. Equipment with low Life Cycle Environmental Impact should be used. Equipment used in all phases, should also be well maintained. Fruits or vegetables and final product cooling equipment should be controlled and well maintained. Low environmental impact transportation means should be preferred. Transportation distance should be minimized where possible. Waste reducing, reusing and recycling should be adopted in all life cycle of the pilot products.

Round 3

Reviewer 4 Report

Dear Authors,

I am sorry to reject the article again.

I made my comments for specific reasons and maybe this was not clear to you.

I explain below pint by point, using the file with your latest reply and part of your answers.

The abstract is very general; please stick to the aims, results and recommendations of your study.

The abstract is a synthesis of your study and should be as specifica s possible to allow interest readers to understand whether the research is on any interest.

How could be your aim “to identify hot spots of agri-production, …” if you do not carry out any contribution analysis?

The results “comprise the size of six impact categories characterization factors”: ok. This doesn’t man anything, as characterization are used to calculate impacts via a given LCIA model

The introduction is very general as well; please focus on the case study areas and provide strong rationale for their selection and representativity. At least you should be able to say what this research adds to the literature; this is critical for article publication.

Being very general, the introduction does not provide enough background for the the specific case studies, including the rationale behind their selection. I understand that your study occurs within a wider research project, however mentioning that does not explain the reason behind cases study selection. Please remember that based on three farm level case studies, making general assertions can be tricky, like this paragraph “The results interpretation emerges the environmental issues of the cultivations and since the method enables in-depth analysis and the selected crops cover a range of products and applied process, it can lead to conclusions and recommendations to develop strategies for improving their ecological performance, able to be extended to the total Balkan Region.”

Then, strengthening the rationale behind case study selection becomes crucial for publishing your study.

Please add a mini-literature review about recent lcas of the selected supply chains and about recent food/agricultural product lcas in the selected countries.

Please give the literature review a dedicated paragraph

The life cycle inventory should be moved to the materials and methods section; emission calculations from the foreground system are missing from the LCI… References are needed per individual data inputs (including primary data) and emission factors.

“The LC Inventory is one of the four phases of the study. That is why it was included in results.”

The goal and scope definition is one of the 4 LCA phases, why is that part not included in the results then? In an LCA, LCI provides data and this phase should be part of the methods and data section, unless the objective of the research is stopping the LCA at the LCI phase. Then, emission factors are not provided, for emissions to soil, air and water. Please detail that part ans especially the model you used to calculate emissions to soil. Those are missing data!

Figure 3: I can’t see the difference between the background and foreground systems; this picture should be included in the methods as well, besides I think one specific picture per product is needed. Not providing data hinders study reproducibility.

Background systems are mentioned in paragraph 2.6. (production of agro-chemicals, fertilizers and machinery and transportation). Ok then, which is the foreground system, the picture does not allow to undestrand that and this figure should be moved to the methods and data seciton as well. PLPease remove figure 1, which is trivial. I am attaching a pdf for you.

In the methods section, the way how characterization factors for impact assessment were used is missing.

Besides, readers should be able to read all figures and tables independently from the manuscript text, then each figure and table shoud detail acronyms

The results section is very poor and contribution analysis is missing, which prevents you to from the hot-spot analysis you anticipated among research aims.

We agree that LCA contribution analysis would help to further and more specific recommendations but this was not prepared for all the three sites. This is a very good remark for further research.

This should be done here, not in further research.

ADDITIONAL COMMENT: please remove any reference to “life cycle analysis” from the manuscript, including the title. The method is an ISO standard and the name in life cycle assessment.

English language is very bad and a revision by a native speaker is compulsory for having your paper published.

Author Response

Dear Reviewer,

thank you very much for your comments and recommendations. Please find below our response in red letters. Also you may check the manuscript with the track changes.

Comment 

The abstract is a synthesis of your study and should be as specific as possible to allow interest readers to understand whether the research is on any interest.

Answer:The abstract has already been modified and I don't really know what else is missing 

Comment : How could be your aim “to identify hot spots of agri-production, …” if you do not carry out any contribution analysis?

Answer: The Inventory analysis and the LCIA leaded to specific observations, that were used for suggestions formation.

Comment :The results “comprise the size of six impact categories characterization factors”: ok. This doesn’t man anything, as characterization are used to calculate impacts via a given LCIA model

Answer: The impact categories characterization factors calculation purpose has been explained in the paper.

Comment :The introduction is very general as well; please focus on the case study areas and provide strong rationale for their selection and representativity. At least you should be able to say what this research adds to the literature; this is critical for article publication.

Being very general, the introduction does not provide enough background for the the specific case studies, including the rationale behind their selection. I understand that your study occurs within a wider research project, however mentioning that does not explain the reason behind cases study selection. Please remember that based on three farm level case studies, making general assertions can be tricky, like this paragraph “The results interpretation emerges the environmental issues of the cultivations and since the method enables in-depth analysis and the selected crops cover a range of products and applied process, it can lead to conclusions and recommendations to develop strategies for improving their ecological performance, able to be extended to the total Balkan Region.”

Then, strengthening the rationale behind case study selection becomes crucial for publishing your study.

Answer: The rationality has been presented in the abstract and the introduction.

Comment: Please add a mini-literature review about recent lcas of the selected supply chains and about recent food/agricultural product lcas in the selected countries.

Please give the literature review a dedicated paragraph

Answer:Two dedicated paragraphs plus a table presenting relevant literature are included.

Comment: The life cycle inventory should be moved to the materials and methods section; emission calculations from the foreground system are missing from the LCI… References are needed per individual data inputs (including primary data) and emission factors.

“The LC Inventory is one of the four phases of the study. That is why it was included in results.”

The goal and scope definition is one of the 4 LCA phases, why is that part not included in the results then? In an LCA, LCI provides data and this phase should be part of the methods and data section, unless the objective of the research is stopping the LCA at the LCI phase. Then, emission factors are not provided, for emissions to soil, air and water. Please detail that part ans especially the model you used to calculate emissions to soil. Those are missing data!

Answer:Goal and scope is a phase where important part of the material and methods for the LCA study preparation is determined. LCI phase in this study is part of the results as data were collected in order to prepare it and it was used for conclusions. Characterization methods include the emission factors and the calculations way (including for emissions to soil) are clearly mentioned and cited.

Comment: Figure 3: I can’t see the difference between the background and foreground systems; this picture should be included in the methods as well, besides I think one specific picture per product is needed. Not providing data hinders study reproducibility.

Answer: Background systems are mentioned in paragraph 2.6. (production of agro-chemicals, fertilizers and machinery and transportation). Ok then, which is the foreground system, the picture does not allow to understand that and this figure should be moved to the methods and data seciton as well. PLPease remove figure 1, which is trivial. I am attaching a pdf for you.

In the methods section, the way how characterization factors for impact assessment were used is missing.

Answer: The impacts characterisation factors that are used are according to cited impact characterization methods which are well known and broadly used. This is mentioned in par. 2.7.

Comment: Besides, readers should be able to read all figures and tables independently from the manuscript text, then each figure and table should detail acronyms

Answer: The acronyms are explained and as they are commonly used in LCA studies the tables and figures can be read and understood independently.

Comment : The results section is very poor and contribution analysis is missing, which prevents you to from the hot-spot analysis you anticipated among research aims.

We agree that LCA contribution analysis would help to further and more specific recommendations but this was not prepared for all the three sites. This is a very good remark for further research.

This should be done here, not in further research.

Answer: It is already considered as a further research opportunity, but impossible to be added in the present paper.

ADDITIONAL COMMENT: please remove any reference to “life cycle analysis” from the manuscript, including the title. The method is an ISO standard and the name in life cycle assessment.

Answer: The “Life cycle analysis” name is also widely used and that “Life Cycle Assessment” is performed is mentioned in the text.

COMMENT English language is very bad and a revision by a native speaker is compulsory for having your paper published.

Answer: An extensive review for English language has been performed.
